# Operational Flood Detection Using Sentinel-1 SAR Data over Large Areas

**Han Cao [1,2], Hong Zhang [1,]\* , Chao Wang [1,2] and Bo Zhang [1]**

[1] Key Laboratory of Digital Earth Science, Institute of Remote Sensing and Digital Earth, Chinese Academy of Sciences, Beijing 100094, China; caohan@radi.ac.cn (H.C.); wangchao@radi.ac.cn (C.W.); zhangbo@radi.ac.cn (B.Z.)

[2] University of Chinese Academy of Sciences, Beijing 100049, China

\* Correspondence: zhanghong@radi.ac.cn; Tel.: +86-10-8217-8186

**Abstract:** Unsupervised flood detection in large areas using Synthetic Aperture Radar (SAR) data always faces the challenge of automatic thresholding, because the histograms of large-scale images are unimodal, which thus makes it difficult to determine the threshold. In this paper, an iteratively multi-scale chessboard segmentation-based tiles selection method is introduced. This method includes a robust search procedure for tiles which obey bimodal Gaussian distribution, and a non-parametric histogram-based thresholding algorithm for thresholds identifying water areas. Then, the thresholds are integrated into the region-growing algorithm to obtain a consistent flood map. In addition, a classification refinement technique using multiresolution segmentation is proposed to address the omission in a heterogeneous flood area caused by water surface roughening due to weather factors (e.g., wind or rain). Experiments on the flooded area of Jialing River on July 2018 using Sentinel-1 images show a high classification accuracy of 99.05% through the validation of Landsat-8 data, indicating the validity of the proposed method.

**Keywords:** synthetic aperture radar (SAR); flood detection; bimodality test; target region search; region-growing

## 1. Introduction

Floods often cover large areas, which are difficult to access and monitor from the ground. Spaceborne remote-sensing data are a well-suited information source used to monitor large-scale flood situations in a time- and cost-efficient manner. Optical satellite imagery, such as those from QuickBird, Landsat, Satellite Pour 1'Observation de la Terre (SPOT), and in particular, the Moderate Resolution Imaging Spectroradiometer (MODIS) have been successfully used in the past to derive water masks of inundation areas [1] due to their straightforward interpretability and rich information content. Systematic flood monitoring using optical imaging instruments during flood events is hampered due to long-lasting periods of precipitation and persistent cloud cover. Spaceborne synthetic aperture radar (SAR) systems are typically powerful tools for near real-time (NRT) flood monitoring due to their near all-weather and all-time capabilities. Most researchers addressing water detection favor working with SAR data, based on sensors such as RADARSAT-1/2, ALOS PALSAR, COSMO-SkyMed, ENVISAT ASAR, TerraSAR-X (TSX), and the recently launched Sentinel-1.

SAR has proven to be an effective tool for water surface detection [2–5], estimation of flood depth [6], and flooding beneath vegetation canopies in certain conditions [7–10]. Considerable efforts have focused on developing algorithms for flood delineation from SAR imagery. Commonly used SAR-based flood extent mapping approaches include simple visual interpretation [11], supervised classification [12,13], image texture algorithms [14], histogram thresholding [15], various multitemporal

change detection methods [16,17], and the active contour models [18,19]. Among these, histogram thresholding is a common approach used to separate flood and non-flood areas in a large area due to the specular backscattering characteristics of active radar pulses on plain water surfaces and the resultant low signal return. For global-scale applications, an automated water-mapping algorithm based on long-term training datasets is introduced to estimate the probability that a pixel is covered by water, given its backscatter and incidence angle [20].

The thresholding method is the most rapid technique for achieving a binary classification of an image. Determining the appropriate threshold has a great influence on the classification results. Typically, because the manual way is time-consuming and not objective, it is not necessary to determine the threshold in this way. Additionally, manual threshold-seeking is known to be inconvenient when water occupies only a small fraction of the image and the distributions of water and background significantly overlap, and often do not exhibit a characteristic bimodal distribution. Rather, the automation of water extraction is necessary. Schumann et al. [15] computed a global threshold value from the radiometric histogram using Otsu's method. The method applies a criterion measure to evaluate the between-class variance of a threshold at a given level computed from a normalized image histogram. The combination of Otsu thresholding and region-growing (RG) has been advocated for determining a suitable threshold for extracting water/flooded areas [21]. However, Lee et al. [22] found that when the object area occupies more than 30% of the whole image, the Otsu threshold approximates the optimal segmentation value. The segmentation performance of Otsu declines quickly when the object area ratio reduces to 10%, which always occurs in large areas in which water occupies a small fraction of space. Thus, further research on automatic and accurate thresholding methods were studied, and completely automated scene-based thresholding approaches have significantly increased in flood detection over the last few years. Among the flood-mapping approaches, automatic thresholding algorithms are usually used to classify SAR data preliminarily. Martinis et al. [23] described an automatic thresholding approach by splitting the scene into subsets or tiles before conducting histogram analysis, and the classification refinement process is a computationally efficient approach that provides reliable results in a rapid mapping context. Recently, an on-demand TerraSAR-X-based flood service was presented [24], which consists of a fully automatic processing chain geared toward NRT pixel-based flood detection using TerraSAR-X data. This processing chain has been adapted to Sentinel-1 (S-1) C-band data [25], operated by the European Space Agency (ESA) in the framework of the European Union's Copernicus Program. In [23], the bimodal tiles selection procedure contain two screening processes, and requires manual input in the second screening procedure; thus, the tiles selection is unstable and has low automation. Schumann et al. [26] examined the potential of remote sensing to monitor flood dynamics in urban areas using a combination of a series of space-borne SAR data (ASAR-WSM, Radarsat-1, ASAR IMG, TerraSAR-X) and aerial photographic images. Matgen et al. [27] proposed a hybrid methodology that combines automatic radiometric thresholding based on gamma distribution curve-fitting and region-growing as an approach enabling automatic, objective, and reliable flood extent extraction from SAR images. However, this method is not fit for images with small water ratios. Chini et al. [28] introduced a hierarchical split-based approach that searched for tiles which were composed of two well-balanced classes (water and non-water) when water class represents only a small fraction of the image. The tiles are selected through parameterization of the distributions of two classes by the Levenberg-Marquardt algorithm whose stability often suffers from the effect of initial weight-value selection; thus, erroneous selections may occur if the tile comprises crop land and high-reflecting urban areas. Thus, it is necessary to develop a time-efficient, fully automatic, and robust bimodal tiles selection procedure when a NRT flooding-detection issue in a large area using SAR is addressed.

Due to the above limitation, a more robust and operational flood detection method is proposed in this work. The method comprises a more robust automatic search technique for regions with clear histogram bimodality, an automatic thresholding approach, and region-growing from a water core to obtain the whole flood area. Seninel-1 scenes, which have a repeat period of 12 days and a resolution of

up to 10 m in interferometric wide swath (IW) mode, were used for flood monitoring. Optical imagery was also used for the validation of flood detection. The remainder of this paper is organized as follows: Section 2 provides a description of the study area and data description. The proposed methodology is described in Section 3. Section 4 is devoted to the experimental results and analysis. Conclusions are drawn at the end of this paper.

## 2. Study Area and Data Description

The SAR image data used for testing the proposed method were acquired during the flood that occurred on the Jialing River in Sichuan Province, China, in July 2018. Sichuan Basin has a subtropical climate with annual rainfall between 900 to 1,200 mm. Heavy rain predominantly occurs in July and August. There are many tributaries of the Jialing River in Sichuan Basin, resulting in frequent flood disasters. The region was subjected to continuous heavy rainfall beginning 2 July 2018, resulting in flooding and significant social and economic losses. On 11 July 2018, the Jialing River in Sichuan Province suffered the largest flood peak in 20 years. The study area is shown in Figure 1. Sentinel-1 SAR images as Level 1 processed ground range detected (GRD) products with interferometric wide swath (IW) mode were collected across the study region. Both pre-flood and flooding images were carefully selected to obtain the permanent water bodies and flooding areas.

One eight-band multispectral Landsat 8 OLI/TIRS image from 16 July 2018 with a resolution of 30 m from the US Geological Survey (http://earthexplorer.usgs.gov/) was used to aid in visual comparison and validation of Sentinel-1 data in flood mapping. The Landsat-8 flooding image was carefully selected based on acquisition time that was closest to that of the SAR flooding data. Although the flooding image from July 16 was severely affected by cloud cover due to precipitation, to obtain the precise flood map reference and validate the effectiveness using SAR data, only a subset area that was cloud-free was chosen for validation, which is discussed in Section 4.2. The detailed characteristics of SAR and auxiliary optical data are provided in Table 1.

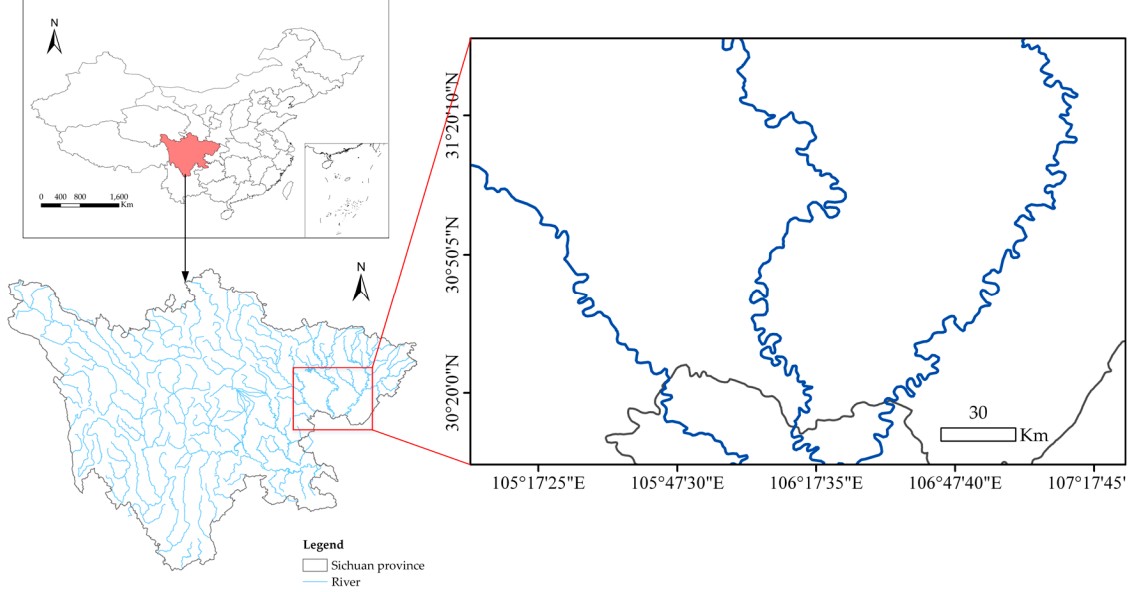

**Figure 1.** The location of the study area.

**Table 1.** Available Synthetic Aperture Radar (SAR) and Landsat-8 data for the flooding event in the study area.

| Sentinel-1 SAR data | | | | | |
|---|---|---|---|---|---|
| **Acquisition Date** | **Mode** | **Orbit** | **Incidence angle** | **Pixel spacing** | **Polarization** |
| 2018-05-04 (pre-flood) | IW | Descending | 30.73°–46.05° | 10 m × 10 m | VV-VH |
| 2018-07-03 (flooding) | IW | Descending | 30.73°–46.05° | 10 m × 10 m | VV-VH |
| 2018-07-15 (flooding) | IW | Descending | 30.73°–46.05° | 10 m × 10 m | VV-VH |
| **Landsat-8 Operational Land Imager (OLI) data** | | | | | |
| **Acquisition Date** | **Path** | **Row** | **Pixel spacing** | **Cloud cover (%)** | |
| 2018-07-16 (flooding) | 128 | 39 | 30 m | 76.56 | |

## 3. Methodology

The procedure proposed in this work consists of three main steps: (1) preprocessing of the pre-flood and flooding images, including calibration, co-registration, and image filtering; (2) water detection of pre-flood and flooding images based on normally distributed SAR data by power transformation, the target region search approach, thresholding, and the region-growing algorithm; (3) post-processing of the water detection map, including multiresolution segmentation-based classification refinement, mountain shadow masking using DEM data and morphological processing, and labeling areas as flood which are classified as "water" in the flooding image and "non-water" in the pre-flood image.

To better explain this procedure, an overview of the methodology for practical flood detection in the large SAR image is illustrated in Figure 2, and the detailed steps are as follows.

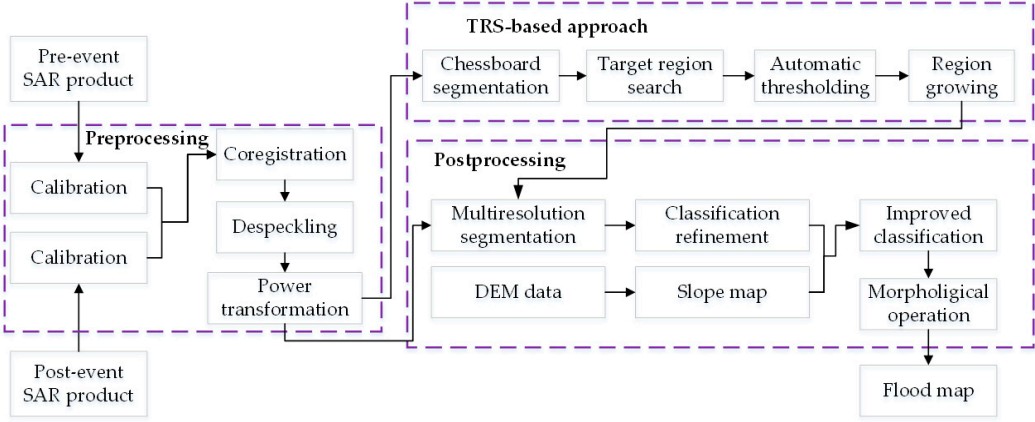

**Figure 2.** The proposed framework for large-scale flood detection. TRS: target region search.

### 3.1. Preprocessing

The Sentinel-1 SAR images were calibrated using the SARscape software. The data calibration compensates for the radiometric influences of different incidence angles (caused by the sensor geometry and topographic characteristics of the surface). To ensure the most exact pixel overlap of the different datasets, the flooding images were registered on the geometry of the early time scene. Then, a 3 × 3 refined Lee filter was applied to all images to reduce speckle noise, which represents a good compromise between the preservation of spatial detail and the signal-to-noise ratio.

### 3.2. Bimodality Test and Transformation of Normality

(1) Bimodality test of normally distributed images

Demirkaya and Asyali [29] proposed, using the maximum normalized between-class variance (BCV), to detect bimodality for different intensity distributions. The between-class variance is expressed as:

$$\sigma_B^2(t) = p_1(t)p_2(t)[m_1(t) - m_2(t)]^2 \tag{1}$$

where $p_1(t)$ and $p_2(t)$ represent the probabilities of the respective classes when the histogram is divided into two classes by the gray-level intensity $t$. $m_1(t)$ and $m_2(t)$ are means of the two classes. The normalized BCV is defined as $B(t) = \frac{\sigma_B^2}{\sigma_T^2}$ and adopts values between zero and one, where $\sigma_T^2$ represents the total variance of the histogram. The maximum of the $B(t)$:

$$B_{max} = \max_{0 \leq t \leq N-1} B(t) \tag{2}$$

can be used as a measure of bimodality. The $B_{max}$ of the bimodal normal distribution was obtained through the simulation of images with two normally distributed regions [29], and it was concluded that images with a $B_{max}$ value less than 0.65 could be assumed as unimodal in the case of normal distributions. It was also used in detecting regions obeying the bimodal generalized Gaussian model in a log-ratio image when the change-detection issue was addressed [30]. In our study, we selected a larger threshold value of 0.75 to detect regions in which a clear bimodal Gaussian model exists, and maximized the probability that the regions contained a considerable proportion of water and other land covers.

(2) Power transformation of the SAR image

Researchers have already established a statistical coherent speckle model for characterizing a speckle as a random variable [31]. The statistical distribution of the multi-look averaging speckle is given by the Gamma distribution in the case of intensity data, and by the multi-convolution of the Rayleigh probability density in the case of amplitude data. Given this, the image intensity in the homogeneous SAR water region can be approximated using a gamma distribution. Kukunage [32] proposed that it is advantageous to convert the gamma distribution to a similar Gaussian distribution by applying a power transformation such as $y = x^v (0 < v < 1)$, where $v$ is the power exponent and is between 0.1 and 0.4 when $x$ obeys the gamma distribution [32]. Ning et al. [33] proved that the Rayleigh distribution can also be transformed into the Gaussian distribution by power transformation when $v$ is approximately at 0.3. Thus, the application of any Gaussian-based analysis algorithm will work well on the power-transformed intensity or amplitude data. In this work, by applying a power transformation to SAR intensity data, the Gaussian bimodality test coefficient $B_{max}$ can be well-applied to the transformed image. $v$ was assigned as 0.1 in our experiments.

### 3.3. Target Region Search (TRS) Approach

The SAR image fails the bimodality test when water occupies a small of fraction of the area in the SAR image and exhibits a unimodal histogram, because the histogram of the water area overlaps with the background. The histogram of the subregions located on the water boundary may be bimodal, with one mode largely representing water and the other representing dry land. The local minimum between the two modes represents the area of transition between water and non-water. These regions are termed "target regions" in this paper. To search for these target regions automatically, a multiscale chessboard segmentation algorithm was introduced to the power-transformed image, which splits the image into square image objects, and then a normal bimodality test was implemented on every segmented region to select the bimodal target regions for further determination of the threshold of water extraction. The proposed target regions search (TRS) approach consisted of the following steps:

(Step 1)　Apply a chessboard segmentation algorithm to the power-transformed image with size s × s, that is, dividing the image into regions of size s × s. The regions in the last row and column are ignored if their sizes are less than s. The total number of regions is denoted by $N$. Let the current region be the first region, that is, $n = 1$.

(Step 2)　Compute the $B_{max}$ value of the current region if $n \leq N$. If $B_{max} > 0.75$, the current region is regarded as a target region; record the location of the current region and the region size. If $n > N$, go to Step 4.

(Step 3)　Let $n = n + 1$ and return to Step 2.

(Step 4)　If at least one target region is searched, stop the process. Otherwise, change the starting point of the segmentation to $(s/3, s/3)$, and repeat Steps 1 through 3. If no target region is found after changing the starting point of the segmentation once, change the starting point again to $(2s/3, 2s/3)$ and repeat Steps 1 through 3. If still no target region is found after changing the starting point twice, go to Step 5.

(Step 5)　Let $s = s - 80$, that is, change the segmentation scale. If $s > 0$, repeat Steps 1 through 4. If $s \leq 0$, stop the process.

Since the shape and area of water in the SAR images are different, for different image scenes, it is impossible to find a uniform region size that can be identified as the balanced mixture of water and background. To ensure that at least one target region can always be automatically searched for different images, the chessboard segmentation was implemented by changing the scale (as shown in Step 5). The initial region size is set to be 480 in Step 1, and the size successively decreases by 80 (e.g., 400, 320, 240, 160, and 80) if still no target region is found after changing the starting point of the segmentation twice under the same region scale. Ultimately, one or more target regions can be selected under a certain region size. These selected target regions all obey bimodal Gaussian distribution, which will be used for a subsequent threshold determination in water extraction.

### 3.4. Thresholding and Region-Growing-Based Water Extraction

The target regions were selected strictly by a high value of $B_{max}$, thus the target regions involved high-contrast intensities and the histogram of the target region value was obviously bimodal, with few superpositions remaining. It is very suitable to use a fast nonparametric histogram-based thresholding algorithm [34]. The histogram of the target region always has more than two peaks and one valley due to speckle noise in SAR images. The desired two peaks and one valley between two peaks can be obtained by smoothing the fluctuations of the original histogram using a Gaussian convolution kernel recursively. For the 1D discrete histogram, the digital Gaussian kernel was used to generate the smoothed histogram. The digital Gaussian kernel with a window size of $W = 3$ is given by $g(-1) = 0.2261$, $g(0) = 0.5478$, $g(1) - 0.2261$. The discrete convolution of histogram $h(t)$ and the digital Gaussian kernel $g(u)$ is defined as [35]:

$$H(t) = \sum_{u=-1}^{u=1} h(t+u) \cdot g(u) \tag{3}$$

The recursive process consists of the following steps:

(Step 1)　Compute the histogram $h_0(t)$ of the selected region, where $t$ is the intensity level of the target region. Let $h_k(t)$ be the histogram after the $k$th iteration of the Gaussian convolution. $k$ is set to 0 at this stage.

(Step 2)　Apply the 1D Gaussian convolution with kernel size 3 to $h_{k-1}(t)$ to derive $h_k(t)$, where $h_{k-1}(t)$ is the histogram after the $(k-1)$th iteration of the Gaussian convolution applying to $h_0(t)$.

(Step 3) Detect the number of peaks of $h_k(t)$. If $h_k(t)$ has more than two peaks, let $k = k + 1$ and return to Step 2. If $h_k(t)$ has two peaks located at $p_1$ and $p_2$, then detect the valley between $p_1$ and $p_2$.

The water mode, detected as the location $p_1$, represents the intensity of water that occurs most frequently. The core water area can be obtained by thresholding the image using the mode value—that is, the pixels with an intensity below the mode are classified as the core water. The global threshold for the core water area is set to the mean of the modes of all the target regions. Finally, a region-growing algorithm (RGA) is used to produce a spatially homogeneous water map to avoid being affected by the speckle noise in the SAR. RGA searches the entire SAR image for pixels adjacent to pixels belonging to a seed region (the core water area, in this case) and that fall within a tolerance criterion.

The tolerance criteria are based on the optimal threshold that segments the water from land. In this study, two thresholding methods were tested to distinguish water from non-water areas. One was the detected valley based on the histograms of target regions, that is, the local minimum (LM) between the two peaks, and the other was the Kitter and Illingworth (KI) algorithm. Since the histograms of target regions can be modeled statically using two 1D Gaussian distributions of the semantic classes, "water" and background, the KI thresholding algorithm [36] was applied for every target region to obtain the threshold, which is based on a minimum error approach to classify the image into the objects and the background. The mean value of the LM thresholds and KI thresholds obtained from all the target regions were regarded as the global LM and KI threshold of the whole image.

*3.5. Postprocessing*

The automatic flood detection method above is pixel-based and can result in the omission of a heterogeneous flooding area. Therefore, the classification refinement technique was proposed by using a multiresolution segmentation algorithm [37]. The image segmentation divides the image into homogeneous polygons, and the percentage of detected water pixels in every segment among the total number of pixels in the segment is computed. Then, the segments that have more than a certain ratio (hereafter denoted "refined ratio") are all identified as water, which can further decrease the omission caused by pixel-based water extraction. Three main parameters, namely, scale, shape, and compactness, are required when using multiresolution segmentation algorithm. The Taguchi optimization method was utilized to select the optimal shape and compactness [38]. Taguchi orthogonal array designs can offer a reduced set of experiments when multiple level factors are designed, which can make the design of experiments easier and more consistent. Different scales and refined ratios are tested in the classification refinement process, and the optimal scale and refined ratio are determined by considering the kappa coefficient, overall accuracy (OA), producer's accuracy (PA), and user's accuracy (UA) of flood detection, which are discussed in Section 4.3.

Another postprocessing step is the removal of false alarms caused by mountain shadow. The shadow effect occurs where the radar wave is blocked from reaching the ground surface, resulting in similar low backscattering as water areas in the SAR, which creates areas of false positives for flooding (error of commission). Of note is the use of the digital elevation model (DEM) to remove the shadow effect, which can reduce the overestimation of flooding. The areas in the flooding map with slope values greater than 15° are considered to be shadows and removed from the result. The DEM based on the Shuttle Radar Topography Mission, released by the Consortium for Spatial Information (SRTMCGIAR-CSI version 4.1) as a freely available global DEM, is introduced for shadow masking. Finally, a median filter with $3 \times 3$ pixel-size kernels were applied to the result to smooth the borders of the flooding map.

## 4. Experiment Results

### 4.1. Application of TRS Approach

The TRS approach was applied to the flooding area of the Sentinel-1 VV polarized image from 15 July 2018, with an image size of 23969 × 36267 (shown in Figure 3a). Since a considerable difference in backscattering exists between the near and far range in SAR images with a wide swath, a unique threshold is not suitable for the whole scene. Using block processing technology, the whole scene was divided into fixed-size non-overlapping blocks (5000 × 5000 in this work, as the yellow dotted grid lines shown in Figure 3a), and the TRS procedure was performed in each block. Only the selected tiles in a block were used to determine the threshold for separating water from land in the current block. If no tile is selected from a block, the threshold for the current block is derived by averaging all available thresholds obtained from its four neighboring blocks. As shown in Figure 3a, there were only two blocks in which no tile was searched—the block at row 1, column 6, and the block at row 1, column 7. The thresholds for separating the two classes in the first block were computed by averaging the thresholds obtained from the block at row 1, column 5 and the block at row 2, column 6, and the threshold for the second block equaled to the threshold obtained from the block at row 2, column 7. The TRS procedure was performed in C++ on a desktop with a 64 b Intel i7-7700HQ 2.8GHz core and 8 GB RAM. A total of 52 target regions were selected in the power-transformed image, as shown in the red boxes in Figure 3a, which are visibly located in the water boundary. The sigma naught ($\sigma^0$) image followed a unimodal long-tailed distribution (see the top graph in Figure 3b) and no threshold could be obtained from the histogram for identifying water, either by visual interpretation or the thresholding methods. The histogram of the power transformed dataset reveals a mixture of two Gaussian distributions (see the bottom figure in Figure 3b), whereas a small prior probability of flood area still results in a near-flat distribution and difficulty in determination of the threshold.

The zoomed subset in the cyan dotted box in Figure 3a is illustrated in Figure 3c with 16 target regions. In the July 16 Landsat scene (Figure 3d), a cloud cover can be observed, which is a clear limitation of optical imagery in flood monitoring. Of the 16 target regions in Figure 3c, four of them were randomly chosen (as the numbered red boxes shown in Figure 3c) for detailed analysis, as shown in Figure 4. From the histogram curves of the target regions, all the target regions appeared to obey a mixture of two Gaussian distributions. The water mode and LM value were obtained by applying the Gaussian kernel smoothing to the histograms, and the KI threshold of the four regions were also computed, as shown in Table 2. Evidently, there was no significant difference between the mode, LM, and KI threshold obtained from the four target regions, and the ranges were 0.0154, 0.0177, and 0.0195, respectively, with none being greater than 0.02. This similarity of the target regions indicates the suitability for subsequent effective threshold detection using the selected target regions.

To validate the robustness of the proposed TRS method, other bimodality test coefficients were compared. Martinis et al. [23] used the coefficient of variation $CV_{X_n}$ (ratio between the mean of a tile to the standard deviation of gray values of the tile) and $R_{X_n}$ (ratio between the mean of a tile to the global intensity mean of the whole image) to select the tiles that contained more than one semantic class using a gray-level SAR image. The bimodality coefficient (BC), which is based on a straightforward empirical relationship between bimodality and the third and fourth statistical moments of a distribution, was applied to the sigma naught in the dB image to select bimodal regions for determining the appropriate threshold that separates flood from non-flood areas [39]. Chini et al. [28] introduced a hierarchical split-based approach that searched for tiles of variable size allowing the parameterization of the distribution of two classes (water and non-water). In their work, a hierarchical tiling of the logarithmic-transformed image was done using a quadtree decomposition, and a sum of two Gaussian curves $h(y) = A_1 e^{\frac{-(y-\mu_1)^2}{2sd_1^2}} + A_2 e^{\frac{-(y-\mu_2)^2}{2sd_2^2}}$ was fitted for values of tiles using the Levenberg-Marquardt (LM) algorithm. Then, the three criteria (hereafter called Chini's method) based on $A_1$, $A_2$, $\mu_1$, $\mu_2$, $sd_1$ and $sd_2$ were tested to evaluate the bimodality of a tile [28].

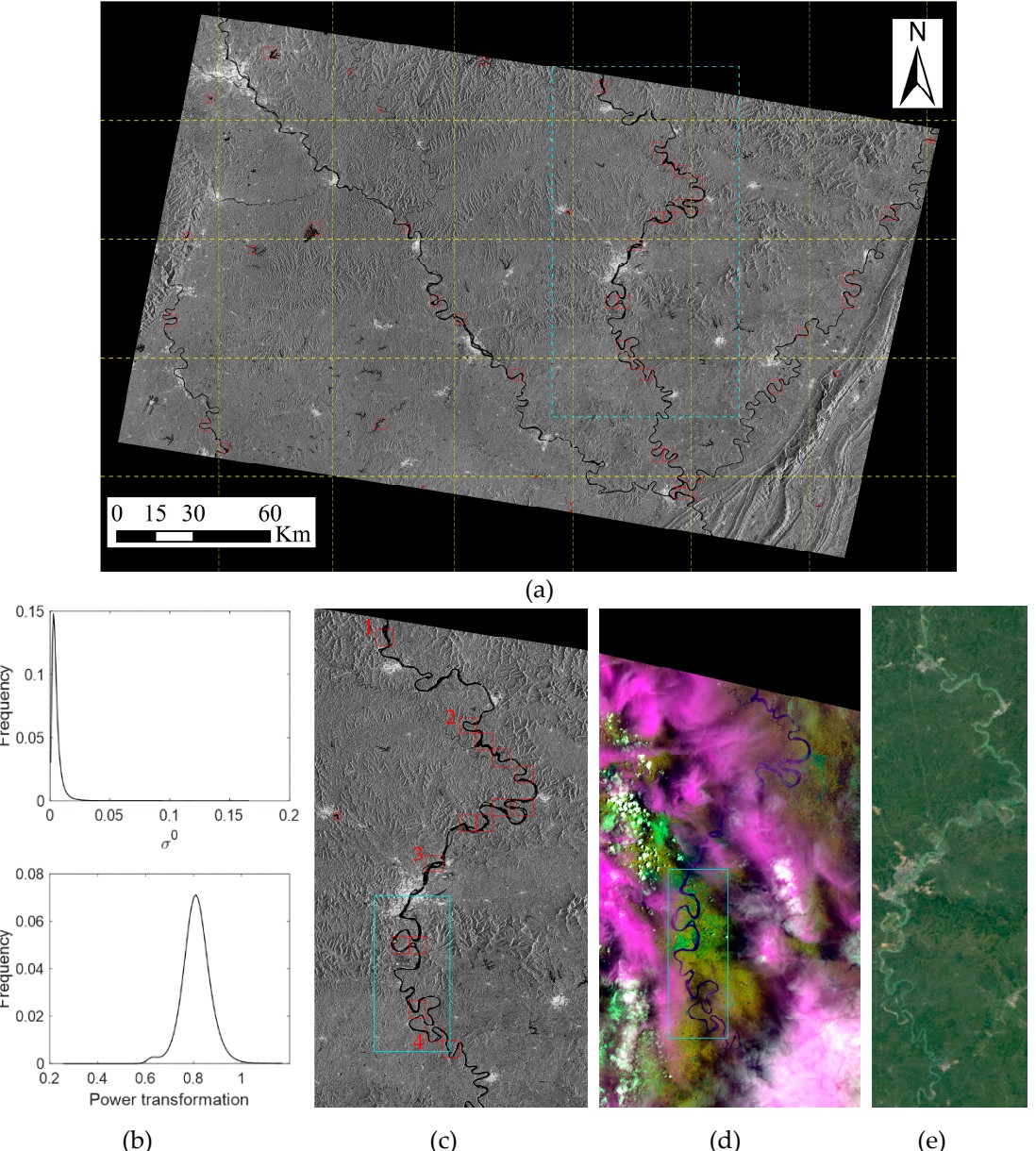

**Figure 3.** (**a**) Sentinel-1 VV polarized image (with size of 23969×36267) of the study area (15 July 2018) with the target regions in red boxes; (**b**) histograms of sigma naught value of full SAR image and the power-transformed image, from top to bottom; (**c**) the zoomed image corresponding to the cyan dotted box in (**a**); (**d**) Landsat-8 OLI/TIRS image (RGB: 5-6-4) of Jialing River (16 July 2018); (**e**) corresponding Google Earth map.

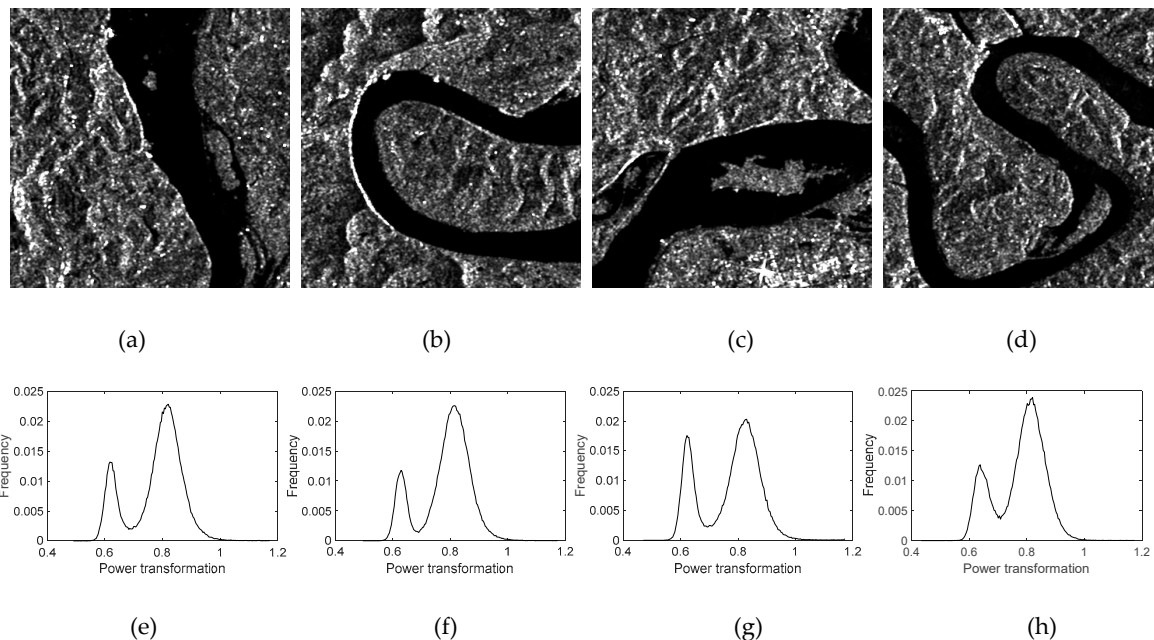

**Figure 4.** (**a**–**d**) The zoomed images of target regions numbered by 1, 2, 3, and 4 in Figure 3c; (**e**–**h**) the corresponding histograms.

**Table 2.** Water mode, LM threshold, and KI threshold of the numbered target regions in Figure 3c.

| Number | Water Mode | LM Threshold | KI Threshold |
|--------|-----------|--------------|--------------|
| 1 | 0.6160 | 0.6914 | 0.6708 |
| 2 | 0.6257 | 0.6939 | 0.6734 |
| 3 | 0.6203 | 0.6909 | 0.6698 |
| 4 | 0.6314 | 0.7086 | 0.6893 |
| Range | 0.0154 | 0.0177 | 0.0195 |

A comparison was conducted between the four bimodality test coefficients, that is, $CV_{X_n}$ and $R_{X_n}$ based on the gray image as proposed in [23], BC based on the sigma naught image as proposed in [39], $B_{max}$ based on the power-transformed image proposed in this paper, and Chini's method based on the logarithmic-transformed image [28]. These three were conducted based on the multiscale chessboard segmentation procedure (as described in Section 3.3), and Chini's method was conducted based on a quadtree decomposition procedure. The entire scene of the S-1 VV polarized image on 15 July 2018 was used to test and compare the four bimodality coefficients. The whole scene contains various water environments, such as the block in row 2, column 5 with the Jialing River crossed, the block in row 2, column 4 with very few water areas distributed, and the block in row 4, column 7 with large topographic relief; thus, the comparison and evaluation of four methods using the whole scene was comprehensive and objective. Figures 5–7 show the tiles' selection results (red boxes) overlaid on the whole scene using $CV_{X_n}$ and $R_{X_n}$, BC, and Chini's method, respectively. Also, the block processing technology was utilized—that is, the tiles selection procedures were performed in each block independently. The selected tiles, which contained a water area, were verified as qualified, and tiles not containing water were considered erroneous through a visual interpretation of the Google Earth map and SAR image. The green-bordered blocks show that there are error selections occurring in the blocks. The false alarms were notated with a yellow triangle beside them. The total numbers of target regions of the whole scene using the four bimodality test coefficients are illustrated in Table 3, as well as the numbers of erroneous tiles and execution time.

The bimodality test using $CV_{X_n}$ and $R_{X_n}$ is the most time-efficient compared with the other three coefficients, but the false alarms occurred in ten blocks. Most of the error selection occurred in the shadow boundary which did not contain a water area. It should be noted that there was a second screening for a subset of all selected regions, which was performed by selecting a number of regions with the lowest 2D Euclidean distance between every selected tile and the mean $CV_{X_n}$ and $R_{X_n}$ of all tiles [23]. However, the number of regions is uncertain and must be manually set by the user. BC performs better than $CV_{X_n}$ and $R_{X_n}$, with errors occurring only in three blocks. The error selection using BC occurs in two scenarios. One is the mixture of high-reflecting urban areas and vegetation land, such as, erroneous tiles in the block at row 3, column 1 in Figure 6. The other is the mixture of a high foreshortening mountain area and low backscattering shadow. The BC-based selected regions are all small in size, which indicates that BC is more sensitive to small regions. The decreasing sizes of tiles used in the TRS procedure makes the BC-based method the most time-consuming. The error selections using Chini's method occur in nine blocks, and the errors occur in the same two scenarios as with the BC-based results. In Chini's work, the threshold for identifying water is determined through the histogram of the union of all selected tiles by reapplying the LM algorithm to the histogram. The union of tiles containing a few error tiles may be bimodal, and optimal threshold can be obtained. When the selected tiles include a number of erroneous tiles, the union of selected tiles may show up to be unimodal or bimodal with small water prior probability, meaning that the selection of the optimal threshold is highly uncertain, such as the block at row 4, column 7 in Figure 7. The erroneous selection of regions results in an erroneous threshold decision and a significant decrease in water detection accuracy.

From Table 3, we can see that the number of selected tiles using the proposed approach is the least. Nevertheless, the target regions can be searched in all blocks except two (see Figure 3a) and the selected tiles are all qualified that contain a distinct number of pixels of water and background whether in plain or complex terrain, which verifies the validity and robustness of the proposed TRS approach. Compared with the quadtree decomposition-based procedure of Chini's method, the only downside of the chessboard segmentation-based method is that it is time-consuming in searching for tiles in the scenario with a small water area, in which case the qualified tiles are often small in size. In the TRS process, the size of the tiles gradually changes from large to small, which wastes a lot of time until the small-sized target regions are searched. However, the advantage of a chessboard segmentation-based tile search is that various sizes of tiles can be set and tried, while the sizes of tiles in the quadtree decomposition-based search procedure are limited to the value of the image size multiplied by $2^{-N}(N$ is a positive integer).

**Table 3.** Bimodal regions obtained using different bimodality test methods in a full SAR scene.

| Method | No. | False No. | Time (min.) |
|---|---|---|---|
| $CV_{X_n}$ and $R_{X_n}$ | 120 | 27 | 8 |
| BC | 546 | 5 | 20 |
| Chini's method | 251 | 27 | 11 |
| $B_{max}$ | 52 | 0 | 15 |

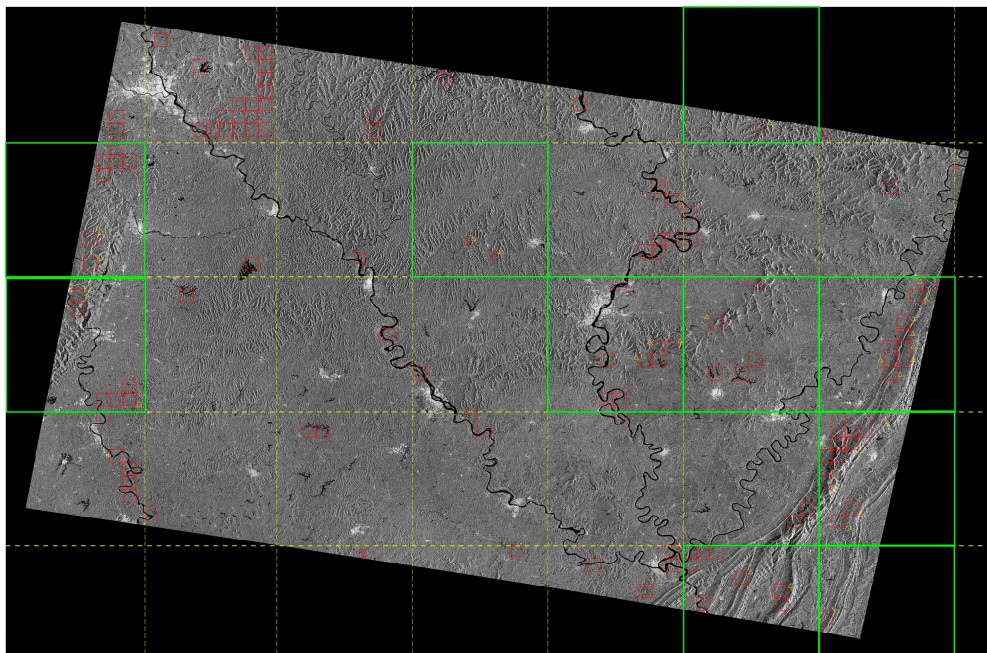

**Figure 5.** Sentinel-1 VV polarized data of the study area (15 July 2018) with the target bimodal regions (in red boxes) using $CV_{X_n}$ and $R_{X_n}$, where the false alarms are notated by a yellow triangle beside them.

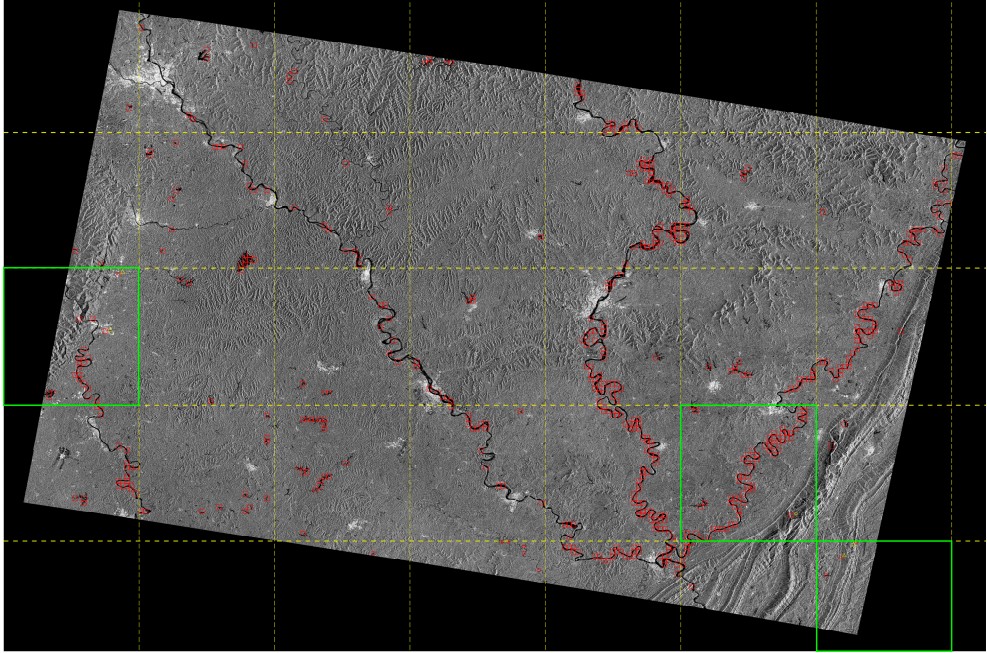

**Figure 6.** Sentinel-1 VV polarized data of the study area (15 July 2018) with the target bimodal regions (in red boxes) using the BC parameter, where the false alarms are notated by the yellow triangle beside them.

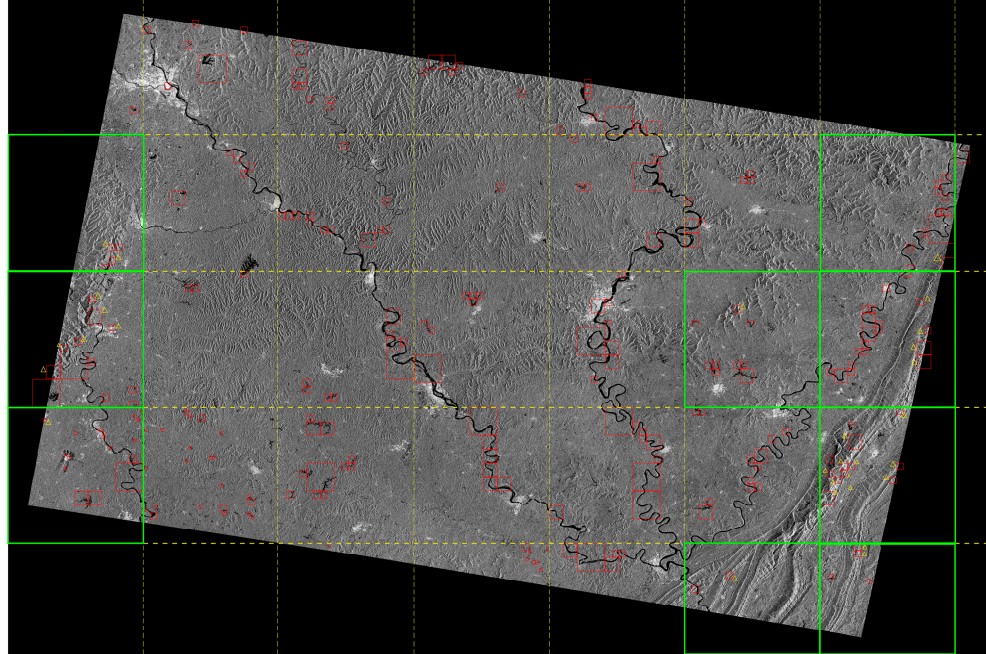

**Figure 7.** Sentinel-1 VV polarized data of the study area (15 July 2018) with the target bimodal regions (in red boxes) using Chini's method, where the false alarms are notated by the yellow triangle beside them.

### 4.2. Flood Detection Comparison by Various Methods in Dual Polarization

To validate the effectiveness of the proposed method in flood detection, a reference flooding map was created for a subset (as shown in the cyan box in Figure 3c) of the Sentinel-1 image based on the water map obtained from the Landsat-8 image. The subset shown in Figure 8 was selected for validation due to the absence of cloud cover in this subset in the Landsat-8 image acquired on 16 July 2018. The Landsat-8 image was first oversampled at the S-1 resolution and then co-registered with the SAR image. The modified normalized difference water index (MNDWI) [40] was applied to the oversampled image, and the Otsu method was applied to the MDNWI dataset to identify areas of water. The reference flooding map (see Figure 8f) was obtained based on visual interpretation of Landsat-8, the MNDWI-derived water map, and the Google Earth map. The subset comprises the heavily flooded Nanchong city region. The two thresholding algorithms were applied to the 15 target regions to obtain thresholds for determining the tolerance value. For every thresholding method, the average value of the thresholds obtained from the fifteen target regions was set as the global threshold for the whole image since there was little difference between the distribution parameters of the regions.

The flood detection in the validation area was implemented based on merging KI thresholding and RGA (hereafter denoted TRS_KI_RGA), and merging LM thresholding and RGA (hereafter denoted TRS_LM_RGA). Other traditional methods, such as global image-based OTSU thresholding (i.e., obtaining the threshold in the whole image using the OTSU algorithm, hereafter denoted OTSU), TRS-based OTSU method (i.e., obtaining the threshold in the target regions using OTSU, hereafter denoted TRS_OTSU), TRS-based KI thresholding (hereafter denoted TRS_KI_THR), and TRS-based LM thresholding (hereafter denoted TRS_LM_THR), were also applied for comparison with the proposed method. Sentinel-1 collects images in VV and VH polarization, both of which have their respective advantages in flood detection. The two polarizations are both used for flood detection for comparison using the same methods, since VV polarization is easily influenced by a change of water surface roughness due to weather factors (e.g., rain or wind). The increasing roughness results in an increasing backscatter return to the SAR, which may not be identified as an inundation area [41].

Figure 8 displays the flood detection map using different polarization without classification refinement (i.e., all the postprocessing steps were done except the result refinement) since our aim was to compare the different methods in water area extraction accuracy, and the subsequent classification refinement will be discussed in the next section. The kappa coefficient, overall classification accuracy (OA), producer's accuracy (PA), and user's accuracy (UA) for the "water" class using different methods under different polarizations were calculated and compared, as illustrated in Tables 4 and 5. OA was computed by dividing the total number of correctly classified pixels (i.e., the sum of the pixels correctly classified as water and non-water) by the total number of reference pixels. PA, also representing the omission errors, results from dividing the number of correctly classified pixels in a water class by the number of reference pixels of the water class. UA, also representing the commission errors, was computed by dividing the number of correctly classified pixels in the water class by the total number of pixels that were classified as water.

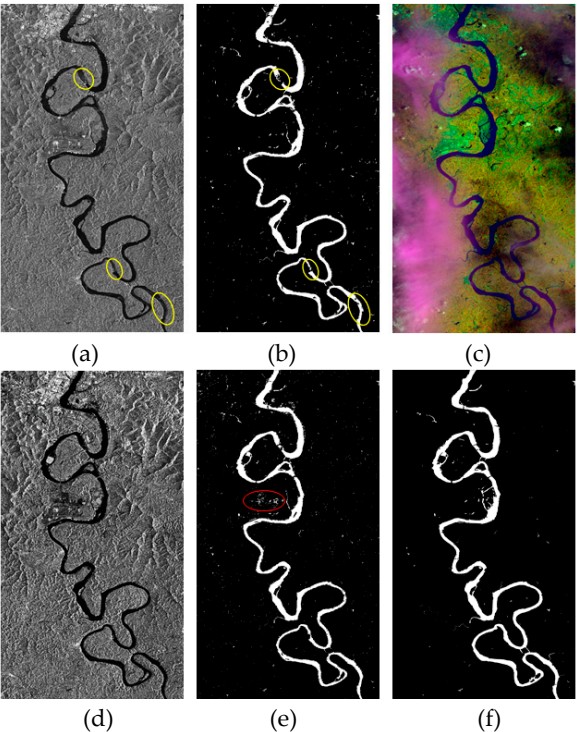

**Figure 8.** Flood detection results in the validation area. (**a**) S-1 VV image acquired on 15 July 2018; (**b**) TRS_LM_RGA-based flood area using the VV image; (**c**) Landsat-8 images (RGB: 5-6-4) acquired on 16 July 2018; (**d**) S-1 VH image acquired on 15 July 2018; (**e**) TRS_KI_RGA-based flood area using the VH image; (**f**) true flooding map obtained from Landsat 8. White: flooding area, Black: non-flooding area.

**Table 4.** Comparison of the accuracy using different methods for VV polarization.

| Method | Kappa Coefficient | OA(%) | UA(%) | PA(%) |
|---|---|---|---|---|
| OTSU | 0.34 | 81.49 | 26.33 | 97.08 |
| TRS_OTSU | 0.72 | 95.54 | 60.89 | 94.74 |
| TRS_KI_THR | 0.88 | 98.60 | 95.34 | 85.29 |
| TRS_LM_THR | 0.87 | 98.42 | 87.41 | 89.39 |
| TRS_KI_RGA | 0.89 | 98.71 | 95.26 | 85.04 |
| TRS_LM_RGA | **0.91** | 98.82 | 93.09 | 89.09 |

*Note*: **Bold** represents the best accuracy for flood detection using VV.

**Table 5.** Comparison of the accuracy using different methods for VH polarization.

| Method | Kappa Coefficient | OA(%) | UA(%) | PA(%) |
|--------|-------------------|-------|-------|-------|
| OTSU | 0.31 | 78.93 | 23.98 | 97.86 |
| TRS_OTSU | 0.63 | 93.53 | 51.07 | 96.25 |
| TRS_KI_THR | **0.89** | 98.59 | 91.08 | 87.72 |
| TRS_LM_THR | 0.82 | 97.95 | 75.57 | 93.17 |
| TRS_KI_RGA | 0.88 | 98.59 | 91.12 | 87.64 |
| TRS_LM_RGA | 0.86 | 98.22 | 82.74 | 92.95 |

*Note*: **Bold** represents the best accuracy for flood detection using VH.

For different polarizations, the whole image-based OTSU thresholding method achieves the worst result (with a kappa coefficient of 0.34 and 0.31 for VV and VH, respectively) since the small prior ratio of water results in an inaccurate threshold for water extraction [22] (the OTSU threshold is always larger than the optimal threshold and a large number of false alarms appear in the detected water area). The OTSU thresholding obtained from the target regions has a smaller threshold than the value obtained from the whole image. Although exhibiting slightly better performance than the whole image-based OTSU method, the target region still results in many false alarms (with a UA of 60.89% and 51.07% for VV and VH, respectively) among the remaining methods.

For VV polarization, TRS_LM_RGA performs best with kappa, OA, UA, and PA, with values of 0.91, 98.82%, 93.09%, and 89.09%, respectively; and the omission error (10.91% in omission rate) is due to inconsistency in the VV backscatter, which may be caused by weather factors, resulting in strong backscattering (highlighted by yellow circles in Figure 8b). Better results were obtained using TRS_LM_RGA (93.09% for UA) than TRS_LM_THR (87.41% for UA), which results in more false alarms due to the water-like pixels in SAR. This indicates that water extraction originating from a seed region can marginally avoid the false alarms due to isolated non-water pixels whose backscatters are a little larger than that of the water core. TRS_LM_RGA performs slightly better than TRS_KI_RGA in PA, which can be explained by the fact that the KI threshold is slightly lower than LM, as shown in Table 2, and the smaller KI threshold results in larger detection omission in the VV image.

The VH can identify a consistent water surface with the kappa coefficient (0.89) in flood detection using the TRS_KI_THR method, which outperforms other methods, although it is characterized by slightly worse performance than the VV in UA. For every method, the VH identifies a greater portion of the region as being flooded compared to the VV, with PA all being larger than that of the VV. This is because cross-polarized data produces a wider range of backscatter values from vegetated land surfaces compared to co-polarized data, leading to a potential overlap with the low backscatter values associated with water and resulting in the misclassification of flooded land [25,41,42]. Compared with the TRS_LM_THR or TRS_LM_RGA, the KI threshold-based methods (TRS_LM_THR and TRS_LM_RGA) performs better in UA using the VH image. The larger LM threshold results in more false alarms in water-like land covers using the VH image.

The results indicate that, using the VV polarization based on the TRG_LM_RGA method, accurate and automatic flood detection is possible utilizing SAR data.

*4.3. Multiresolution Segmentation-Based Refinement*

In this study, the Taguchi optimization method was used to select the best combination of shape and compactness to segment the Sentinel-1 image, as discussed in [38]. Five values of each parameter [38] were used, and the L25 array design was selected. The best levels of parameters were then determined by calculating the signal-to-noise ratios (SNRs). The total variance of segments was used to calculate the SNR, that is, $\text{SNR} = -10Log_{10}\left(\frac{\sum_{i=1}^{N} n_i Var_i}{\sum_{i=1}^{N} n_i}\right)$, where $N$ is the number of segments, $n_i$ is the number of pixels in the $i$th segment, and $Var_i$ is the variance of the $i$th segment. The optimal shape and compactness were obtained by maximizing the SNR. By conducting 25 experiments, the

best value for shape and compactness was 0.1 and 0.5, respectively. Under the selected shape and compactness, the scale parameter was determined through the TRS_LM_RGA-based classification refinement experiments using the VV image, setting the scale to 30, 40, . . . , 120 and setting the refined ratio to 0.5, 0.6, and 0.7. Too small a value of the refined ratio easily results in false alarms, while too large a value will not significantly enhance the accuracy. The kappa coefficient, OA, UA, and PA curves on different scales and refined ratios are illustrated in Figure 9, which reveal a significant increase in the kappa coefficient, OA, and PA, and a decrease for UA. The UA decreases less than one percent when the refined ratio is 0.7, whereas the PA increases by more than four percent. A trade-off is considered between UA and PA. The refined ratio is set to be 0.7 while the reduction in UA is minimized (92.37% in UA) under the refined ratio, and the scale is set to be 70 while the kappa (0.9250), OA (99.05%), and PA (93.66%) all reach maximum values under the scale when the refined ratio is set to 0.7.

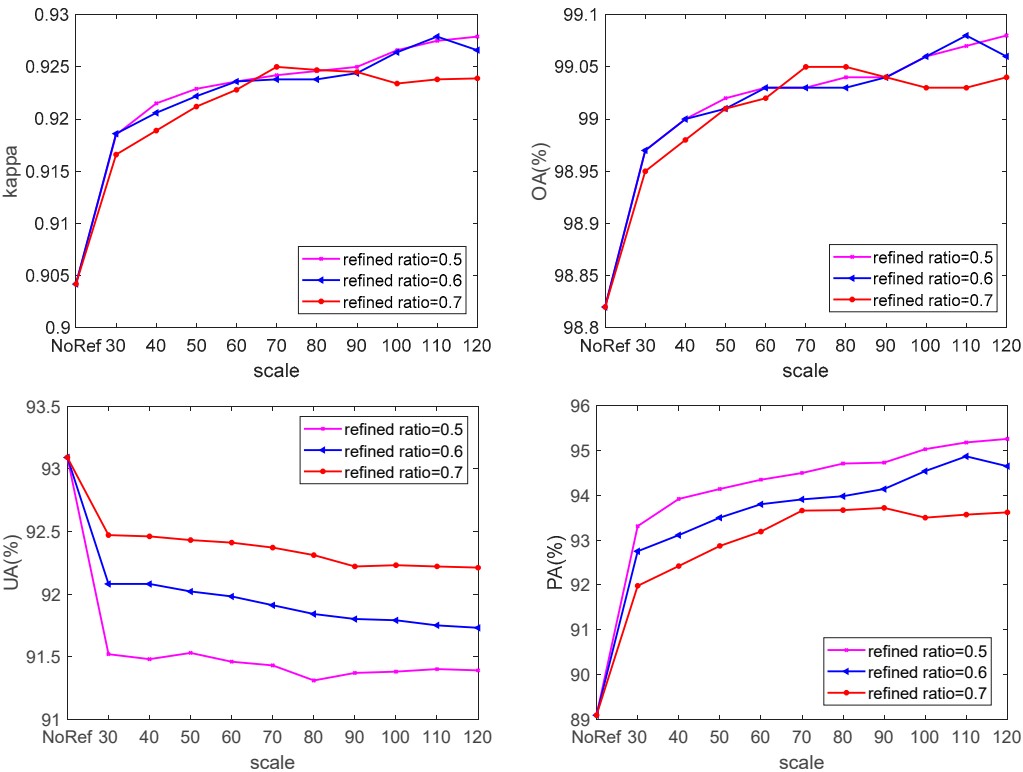

**Figure 9.** TRS_LM_RGA-based classification refinement results using the VV image under different scales and refined ratios. NoRef refers to water detection without classification refinement.

For the selected scale, shape, compactness, and refined ratio, the segmentation was applied in the VV image, and the TRS_LM_RGA-based result (Figure 8b) was refined as displayed in Figure 8. The classification refinement lies in the heterogeneous flood areas which may be caused by wind, as shown by the three regions of interest (ROIs) in yellow boxes in Figure 10c. An omission occurred in flood detection using the TRS_LM_RGA method in the VV image, as shown by the red circles in Figure 10f. Through classification refinement, the omission areas can be classified as flooding (see Figure 10h), which improved the flood detection accuracy significantly. The segmentation generally does not work well on a small water area; thus, the object-oriented water extraction always leads to omission in small ponds. The refinement applied on the result obtained from a pixel-based method can enhance the consistence of large rivers and not ignore small water areas.

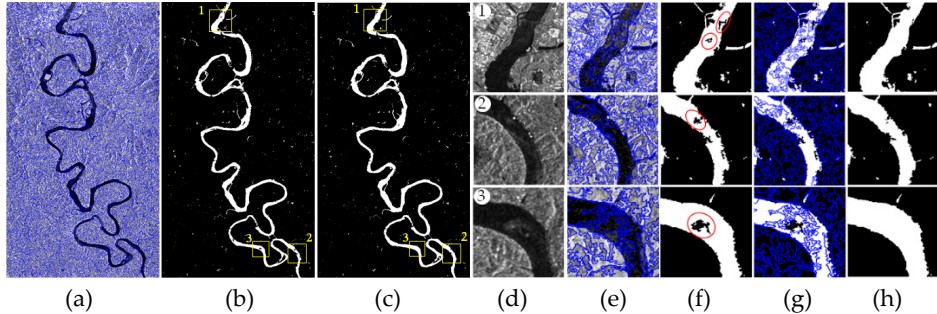

|  |  |  |  |  |  |  |  |
|:---:|:---:|:---:|:---:|:---:|:---:|:---:|:---:|
| (a) | (b) | (c) | (d) | (e) | (f) | (g) | (h) |

**Figure 10.** Classification refinement of TRS_LM_RGA-based result using the VV image. (**a**) multiresolution segmentation result of the VV image with scale = 70, shape = 0.1, and compactness = 0.5; (**b**) TRS_LM_RG-based result without classification refinement; (**c**) TRS_LM_RGA-based result with classification refinement; (**d**) zoomed ROIs in (**c**); (**e**) segmentation results; (**f**) flood maps without refinement; (**g**) segmentation overlaid on the flood maps without refinement; (**h**) flood maps after refinement.

## 4.4. Flood Dynamics

The multiple Sentinel-1 images of the study area over the pre-flood and flooding period enabled the monitoring of flood dynamics. Firstly, the permanent water area was detected using the pre-flood SAR image on 4 May 2018 by the proposed method, as well as the water extent in the flooding period on 3 and 15 July 2018. Then, the flood extents on 3 and 15 July 2018 were obtained by removing the permanent water extent. Figure 11a displays the flood dynamics between the two flooding dates overlaid on the pre-flood SAR image, containing three classes, that is, the flooded area at both dates (in red), the flooded area on 3 July, and that which receded on 15 July (in blue), and the new flooded area between 3 July and 15 July (in yellow).

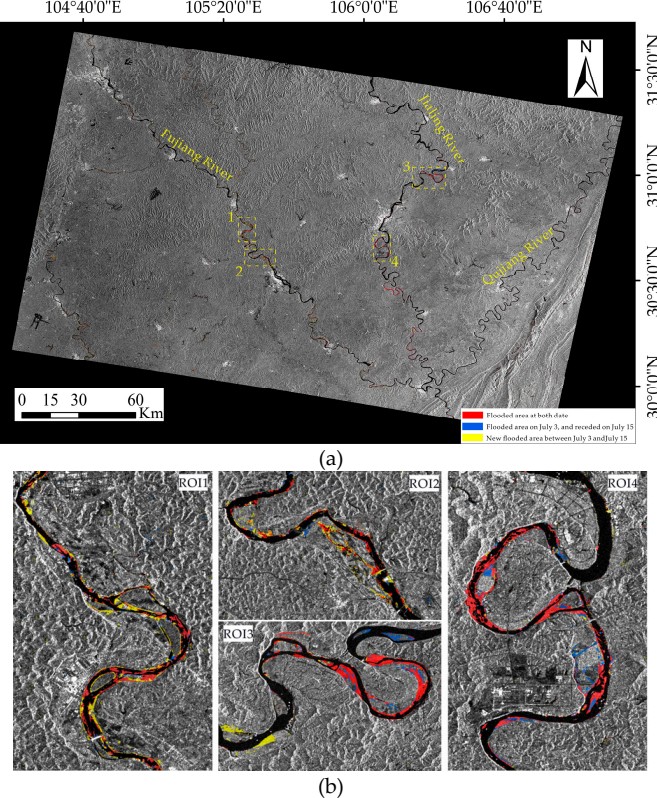

**Figure 11.** (**a**) Flood dynamics for the study area on 3 and 15 July 2018 overlaid on the pre-flood SAR image on 4 May 2018; (**b**) four ROIs selected in correspondence to the numbered yellow-dotted rectangles.

Fujiang River—as a tribute of the Jialing River—and the main stream of Jialing River both suffered from heavy flooding more than the Qujiang River, another tribute of Jialing River. Multiple sectors along the Fujiang and Jialing Rivers were kept inundated throughout the whole study period, such as ROI1, ROI2, ROI3, and ROI4 shown in Figure 11b. Only a few flooded areas on the Jialing River beach receded (blue parts in ROI3 and ROI4), and a few new flooded areas appeared in the vegetation field next to the main stream (yellow part in ROI3) between 3 July and 15 July. Fujiang River suffered a flood peak on 11 July, and some new flooded areas appeared and did not recede on 15 July (as the yellow parts shown in ROI1 and ROI2). The analysis of flood dynamics using flooding images reveals significant information for flood management and monitoring of the study area.

## 5. Discussion

The proposed TRS-based approach splits a power-transformed SAR image to segments using the chessboard segmentation method, in which the segments obeying a clear bimodal Gaussian distribution are searched by a bimodality test coefficient. In our method, the segmentation starting-point and scale gradually changes until the qualified tiles are selected, so that very small water areas can be detected. Instead of using time-consuming parametric histogram-based thresholding algorithms, nonparametric histogram-based thresholding based on a recursive Gaussian convolution kernel was applied to the histograms of selected target regions to obtain a threshold for separating water from land, which is robust and computationally fast for time-critical applications (e.g., flooding monitoring). The robustness of the TRS method renders the accurate determination of thresholding parameters in water detection in multiple kinds of terrains.

Aiming at the flood mapping of the full SAR scene, the block processing technology is utilized—that is, only the selected tiles from a block are used for determining the threshold of water detection in the current block. By using optimal thresholds suitable for water detection in each block, the precise flood mapping of each block in the whole SAR scene can be automatically and feasibly obtained without a refinement process, which meets the application requirements of near-real-time flood monitoring. Thus, when there is a considerable contrast between water and other land covers, and water areas are highly homogeneous, the proposed procedure without result refinement provides an opportunity for efficient, automatic, and robust flood detection over large areas using SAR data with wide swath.

Furthermore, with the refinement of the results, a more precise flooding map can be obtained, especially in waters with strong heterogeneous scattering. However, the multiresolution segmentation-based classification refinement is time-consuming and non-automatic. The parameters of scale and ratio in the refinement process are optimized by using ground truth, and may not be the optimal parameters when using other SAR images. Future work will focus on adaptive parameterization, which is suitable for various SAR data.

## 6. Conclusions

In this paper, an operational and robust unsupervised framework for flood detection using Sentinel-1 SAR images in large areas has been proposed. The method is capable of obtaining the target regions that comprise adequate portions of flood and non-flood areas through applying power transformation to images for data obeying the Gaussian distribution, multiscale chessboard segmentation of images, and bimodality tests. Two thresholding strategies (KI and LM thresholding) combined with a region-growing algorithm for flood extraction were applied to selected target regions, and then compared. Moreover, a novel classification refinement technique was introduced using multiresolution segmentation to obtain a more precise flooding map.

The proposed TRS chain has the strongest robustness compared to other bimodality tests in the full SAR scene, which contained different water environments and terrains. Based on the reference map derived from the Landsat image, compared to the KI-based region-growing method, the LM threshold-based region-growing algorithm performed best in flood mapping with an OA of 98.82%. Also, compared with VH data, VV polarization data were found to be more suitable for accurate flood

detection. The segmentation-based classification refinement can address the omission in heterogeneous flooded areas with high backscattering caused by weather factors (e.g., wind or rain), and improve the accuracy of flood mapping to 99.05%. These findings indicate that accurate flood detection over large areas is possible, and the technique can facilitate the use of SAR data with wide swath in the operational applications of water resource management over large areas.

**Author Contributions:** H.C. conceived and performed the experiments; H.Z. supervised and designed the research and contributed to the article's organization; C.W. and B.Z. carried on the result analysis. H.C. and H.Z. drafted the manuscript, which was revised by all authors. All authors read and approved the final manuscript.

**Funding:** This work was supported by the National Key Research and Development Program of China (2016YFB0501501) and the National Natural Science Foundation of China (Nos. 41331176, 41371352 and 41401514).

**Acknowledgments:** The authors would like to thank ESA for providing the Sentinel-1A SAR data and USGS for providing Landsat-8 data.

**Conflicts of Interest:** The authors declare no conflict of interest.

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
