# Peer review of "Operational Flood Detection Using Sentinel-1 SAR Data over Large Areas"

_water, doi:10.3390/w11040786_

Round 1
Reviewer 1 Report
This study using Sentinel-1 SAR to detect flooded area looks promising. The methodology seems appropriate, and the results are well presented. Abstract needs to be re-written. Much improvement in writing is needed. Here are very few examples:
Line 28: replace regions with areas
Line 37: Replace nearly with near
Line 47: Remove always
Line 50: data sets should be a single word i.e. dataset
Line 54-55: “It is time…..monitoring” Restructure this sentence or join to another sentence.
Line 56: Replace the word ‘hampered’ with something like difficult or inconvenient
Line 66-68: Restructure this sentence
Please also include uncertainties and limitations of the methods used.
Author Response
Dear Editor 18 February , 2019
Dear Reviewers
Manuscript ID: water-433441 entitled
“Operational Flood Detection Using Sentinel-1 SAR Data Over Large Areas”.
We appreciate the thorough reviews provided by the reviewers and the handling editor. We agree with these suggestions, supplement the comparative experiments, and have revised the manuscript accordingly. Below is our response to their comments. We hope these revisions resolve the problems pointed out by the reviewers. In the revised manuscript and this file, the red, magenta, and blue parts are revisions suggested by three reviewers, respectively.
Regards,
Hong Zhang
zhanghong@radi.ac.cn

Reviewer 2 Report
I read the manuscript by Han Cao et al. entitled " Operational Flood Detection Using Sentinel-1 SAR Data Over Large Areas ", with a great interest.
From my point of view, this work is very interesting and represent a good contribution to the research in the field of flood mapping. Authors propose an interesting method for river detection using high-resolution satellite SAR image.
The paper is well written although the quality and the clarity slightly decrease from the introduction and method part to the result part where sentences becomes sometimes harsher to follow and understand. I then recommend the authors to try to make the last parts of the paper slightly easier to read.
However, I have four main concerns on the methodology and the set up of the experiment.
The first is that authors missed an important paper in the literature that shares a lot of similarities with their work. Indeed Chini et al (2017) proposed a hierarchical tiling method for mapping a class of interest (flooded area as an example) that only covers a small area of a large SAR image.
M. Chini, R. Hostache, L. Giustarini and P. Matgen, A Hierarchical Split-Based Approach for Parametric Thresholding of SAR Images: Flood Inundation as a Test Case, IEEE Transactions on Geoscience and Remote Sensing, 55(12):6975-6988, 2017.
To do so, they iteratively split the image and seek detect subtiles of various size for which the histogram is bimodal. In this context, Authors can’t avoid to make reference to this method, in addition to the one proposed by Martinis et al. (2009) as an other state of this art method sharing the same objective. As a matter of fact, I would recommend the authors to better replace their contribution in the literature and highlight the originalities and innovations of their method.
The second is about the ground truth generation and the not completely fair comparison with other methods. The ground truth is generated from a cloudy landsat image of spatial resolution lower than that of the SAR image using a thresholding on NDWI. However, no detail is given on how this threshold is set and no real comment on the limitations of this kind of “ground truth” is provided. Moreover, the comparison with other methods do not use the same input data: the original image for the first one, the calibrated image for the second one, and the power transformed image for the proposed method. This can of course add some bias. So I suggest that the authors either use the same input image or justify the use of various input images for the different method. Moreover, authors only mentioned a few methods. Maybe other could be cited. In this respect, authors can find alternative methods in Landyut et al (2019).
L. Landuyt, A. Wesemael, G. J.-P. Schumann, R. Hostache, N. E. C. Verhoest, and F. M. B. Van Coillie, Synthetic Aperture Radar based flood mapping: an assessment of established approaches, IEEE Transactions on Geoscience and Remote Sensing, 57(2):722–739, 2019
The third one relates to the input image that is used for the experiment. Indeed, it seems to me, if I am not wrong, that the method is tested on a subset of a sentinel image. I am consequently wondering why starting from a subset centered on the riverstream while the objective of the method is to proposed a method that work for large images where water covers only a limited area. In my opinion, using the full SAR scene would give more evidence that the method is efficient and accurate.
The last one is more specific and related to the power transformation that is proposed in section 3.2. Usually, when processing SAR images, a log transformation is applied especially because this helps in making water distribution closer to a Gaussian. Consequently, I am wondering why authors apply a power transformation that is to my knowledge not the common practice with SAR data. Could the author please justify this choice?
Please find hereafter some more specific remarks.
1. Line 171: It is not clear to me what is a “chessboard segmentation algorithm”. Is this expression standard or is it the name you choose for your method. Is the selection of bimodal tiles a “segmentation” ?
2. Line 204: Could you please clarify what is hk-1
3. Line 213: could you please explain ou RGA helps in not being affected by speckle.
4. Line 225: The method proposed above makes use of spatial information through the RGA. Could the authors please clarify?
5. Line 228: the word “segment” is maybe not appropriate (for me a segment is a finite part of a line), maybe polygon could be an alternative.
6. Line 233/ Could you please write a rapid description of the Tagushi method ?
7. Lines 234-237: Here the parameters of scale and ratio are optimized using the ground truth. How this could then be applied to other SAR images where no ground truth is available. This is contradiction with the earlier statement that the method is automatic and unsupervised.
8. Line 257: “no decision threshold can be obtained”: it is not completely true as one can then detect a valley point in the histogram (even if this point is not the optimal threshold).
9. Line 246: “a morphological operator” is far too vague in my opinion. Could you please specify (many morphological operator exist)?
10. Line 297: I would suggest to replace “can search all correct regions” by find only correct regions.
11. Line 300: I do not understand the meaning of “most error selection”. Could the authors please clarify?
12. Line 346: wind does not always occur during flood events
13. Line 355: What is meant by “true” threshold? Optimal? Could you please clarify?
14. Line 380-381: I do not know how authors justify this sentence.
15. Line 383: maybe “lower” instead of “less” and “larger” instead of “more detection”.
16. Line 426: please correct the end of the sentence
17. Line 449-450: For me it seems difficult to claim that these areas correspond to new flooded areas.
18. Lines 456-457: the algorithm does not use multitemporal images. It is applied sequentially to different images which is completely different (see Change detection for instance). Could you please correct?
19. Figure 9: the quality of this figure should be improved.
20. Line 459: the “power transformation of images to normal distribution”: please clarify this
21. Line 465: please remove “coefficients”
22. Line 470: how do you determine if a higher basckatter is du to wind or to emerging ground (e.g. island)?
23. Conclusion: Too light and not really well written and clear in my opinion. The main findings, with the numerical values should be better reported. Could you please improve this section?
Author Response
Dear Editor 18 February, 2019
Dear Reviewers
Manuscript ID: water-433441 entitled
“Operational Flood Detection Using Sentinel-1 SAR Data Over Large Areas”.
We appreciate the thorough reviews provided by the reviewers and the handling editor. We agree with these suggestions, supplement the comparative experiments, and have revised the manuscript accordingly. Below is our response to their comments. We hope these revisions resolve the problems pointed out by the reviewers. In the revised manuscript and this file, the red, magenta, and blue parts are revisions suggested by three reviewers, respectively.
Regards,
Hong Zhang
zhanghong@radi.ac.cn

Reviewer 3 Report
The research question is around the topic of introducing a robust method to detect flood water in SAR-derived flood imagery, where water is a small fraction of a much larger scene. The manuscript introduces a semi-automated processing chain for monitoring large-scale flooding in near-real time. The methodology involves steps for pre-processing of the SAR product, then a Target Region Search algorithm (segmentation, histogram thresholding and region growing) and finally a post-processing phase of image segmentation, classification refinement and change detection (based on a pre-flood image and DEM data). The method was applied to two Sentinel-1 SAR data products and, in comparison with a LandSAT optical satellite image of the same flood, show that flood pixels can be detected with up to 99% accuracy with this approach.
The concept of the paper is sound and would be an interesting contribution to the field of automated flood detection algorithms using SAR satellite data if a small number of issues could be addressed:
Moderate issues:
- I feel that the research question/ aim of the manuscript is not 100% clear. As automated frameworks for flood detection from SAR data is not new, please include at least one sentence that tells is more about what is new, unique or innovative. It would be very useful to explain clearly why this work was carried out and what sets it apart from previous work done.
- The TRS-based approach is an interesting concept put forward in this manuscript. So it would be good to review more of the literature on the idea of splitting the SAR image into sub-images/ tiles, compare it with methods introduced in similar manuscripts (e.g. Chini et al., 2017 – reference given below), and if appropriate highlight how TRS is different from what has been done before.
- In my opinion, the structure of the results section needs some revision. I found it difficult to follow sections 4.2 and 4.3 because some paragraphs did not flow together (e.g. line 343) and certain results and discussion seem to be outside of the original scope (again following figure 2 and section 3). In particular, it is not clear why the results section should look at differences in SAR VV and VH polarization (this was not indicated in the methodology).
- The manuscript uses Overall Accuracy to validate the final result, but OA is not introduced. What is it and why was it chosen here (along with PA and UA)?
Minor issues:
- Figure 2 has not included the histogram/thresholding step in the methodology. Is this deliberate?
- The word ‘different’ is ambiguous and a little confusing in the headers 3.4 and 4.2, would the authors consider reviewing the wording?
- The text of the main manuscript is in good English. But I would suggest some review of the Abstract, particularly the first sentence, which is not clear.
- Is line 148 ‘adopys’ supposed to say adopts?
- Would it be possible to improve the resolution of the figures as it is sometimes difficult to read the text? E.g. in Figure 1 I cannot read the legend or scale bar, in Figure 5 I cannot read the graphs or the text in Figure 5a. In Figure 7 the text appears blurred. The same comment applies to the text boxes/numbers in Figure 5a.
- It might be a good idea to choose a different color scheme for the small boxes in Figures 3 and 5. They disappear when the manuscript is printed in black and white.
- There are a few variables / terms / abbreviations which have not been introduced / explained in first use, such as IW (line 92), v (line 160), g (line 202), KI algorithm (line 218), Sigma naught data/image (line 253) and ROI (line 420). Could this be done?
------------------
Chini, M., Hostache, R., Giustarini, L. and Matgen, P., 2017. A hierarchical split-based approach for parametric thresholding of SAR images: Flood inundation as a test case. IEEE Transactions on Geoscience and Remote Sensing, 55(12), pp.6975-6988.
Author Response

(The authors gave the same response as above.)

Round 2
Reviewer 1 Report
Good improvement in the manuscript
Author Response
Dear Editor
Dear Reviewers
Manuscript ID: water-433441 entitled
“Operational Flood Detection Using Sentinel-1 SAR Data Over Large Areas”.
We appreciate the thorough reviews provided by the reviewers and the handling editor. We agree with these suggestions, and have supplemented the comparative experiments using the whole scene of SAR data, and have revised the manuscript accordingly. Below is our response to their comments. We hope these revisions resolve the problems pointed out by the reviewers. In the revised manuscript and this file, the red, magenta, and blue parts are revisions suggested by three reviewers, respectively.
Regards,
Hong Zhang
zhanghong@radi.ac.cn
Reviewer 2 Report
I read again the manuscript by Han Cao et al. entitled " Operational Flood Detection Using Sentinel-1 SAR Data Over Large Areas ", with a great interest.
From my point of view, this work is very interesting and represent a good contribution to the research in the field of flood mapping. Authors propose an interesting method for river detection using high-resolution satellite SAR image.
I guess Authors made the revision in a hurry as the updated text has often errors and sometimes suffer from a lack of clarity. The English should be polished and proofed again.
From a scientific point of view, I believe that authors have significantly improved the paper and they address the majority of the points raised by the reviewers, but some points are still debatable.
As a matter of fact, I think the paper is not acceptable unless revisions are made.
As I already mentioned in the first round of the review process, the method that is proposed has strong potential but the paper is not convincing enough. Indeed, Authors argue that the proposed methods allow to handle large images where water only covers a small portion which I thing is true.
But unfortunately, they only apply the method on a subset of the sentinel image centered on the river. Moreover, the comparison between the proposed methods with others is carried out on selected, rather small, subareas which is not convincing. I have two concerns with this:
1. This does not clearly show the method capability for a large image
2. The selection of subareas (fig 5) is not motivated and could be responsible for an unfair comparison.
With that respect, I really encourage Authors to present results obtained from the full Sentinel scene processing. Of course a discussion based on more local areas is possible. Moreover, Authors should explain which method they used to determine false alarm in table 3.
Another important point is the necessity of significantly improving figures 5 as this is important for the paper. First, the colored of the boxes are not distinguishable enough. Second I did not find any blue square in fig 5b and c which is a problem as table 3 claims that some 1 tile is selected using Bmax for these two subtiles. Next, the numbers are not visible in fig 5c. Again Why did the author choose this specific small areas? Moreover I do not see the point of the red squares that add complexity.
Please find hereafter some more specific remarks.
1. Figure 4. One can’t see the number anywhere (“numbered target regions”)
2. All over the manuscript, could the authors please proofread. The English writing should be improved. There are mistakes especially in the added text
Author Response

(The authors gave the same response as above.)

Reviewer 3 Report
Thank you for the revisions made to the manuscript. The intensions of your work are now more clear.
One last minor point - I believe you still need to explain that you use PDF in your histogram plots, rather than the original histogram data. And there are also a few spelling errors which should have been picked up in the editing stage. Nevertheless, I believe the concept of the paper is sound and would be an interesting contribution to the field of automated flood detection algorithms using SAR satellite data.
Author Response
Dear Editor
Dear Reviewers
Manuscript ID: water-433441 entitled
“Operational Flood Detection Using Sentinel-1 SAR Data Over Large Areas”.
We appreciate the thorough reviews provided by the reviewers and the handling editor. We agree with these suggestions, and have revised the manuscript accordingly. Below is our response to their comments. We hope these revisions resolve the problems pointed out by the reviewers. In the revised manuscript and this file, the red, magenta, and blue parts are revisions suggested by three reviewers, respectively.
Regards,
Hong Zhang
zhanghong@radi.ac.cn
